

# Research progress on the predictive role of sarcopenia in the course and prognosis of inflammatory bowel disease

Yang Liu[1] and Linglin Tian[2]

[1] Shanxi Medical University, Taiyuan, Shanxi, China
[2] Department of Gastroenterology, The First Hospital of Shanxi Medical University, Taiyuan, Shanxi, China

## ABSTRACT

Sarcopenia is a syndrome characterized by a progressive and extensive decrease in skeletal muscle quality and function. With the development of imaging technology in recent years, the understanding and research on the pathogenesis, diagnosis, and evaluation of sarcopenia have gradually improved. More than one-third of patients with inflammatory bowel disease (IBD) have sarcopenia (*Ryan et al., 2019*), and as a new and unique body composition evaluation index, it is critical for predicting the clinical course, prognosis and postoperative complications of patients with IBD. However, there are limited research summarizing the prevalence of sarcopenia among IBD. Furthermore, there is a scarcity of studies establishing the precise criteria for diagnosing sarcopenia in these patients. This article summarizes the evaluation of sarcopenia and its recent advancements in predicting the course and prognosis of IBD.

## INTRODUCTION

IBD is a group of chronic nonspecific intestinal inflammatory disorders of unknown etiology characterized by recurrent abdominal pain, bloody mucoid stool, weight loss, and anemia. The three primary types of IBD are Crohn's disease (CD), ulcerative colitis (UC), and indeterminate colitis (IC). Rapid industrialization, changes in dietary habits and lifestyle are all factors affecting the incidence of IBD, increasing the number of IBD patients each year (*Le Berre, Honap & Peyrin-Biroulet, 2023*). No treatment or gold standard for the diagnosis of IBD is currently available. Surgical intervention is necessary when medical therapy is ineffective, especially for patients with CD who may require multiple surgeries. This significantly increases the medical burden and deteriorates patients' life quality.

Sarcopenia is prevalent among patients with IBD, and its pathology may be influenced by disease severity and medications used for treatment (*Bryant et al., 2013*; *Zhang et al., 2017a*; *Zhang et al., 2017b*). Moreover, Sarcopenia can predict inflammatory status and surgical outcomes (*Campbell et al., 2022*); therefore, it is vital for the identification of high-risk patients with poor surgical prognosis (*Ryan et al., 2019*). IBD patients are prone to malnutrition during the inflammatory process due to various factors, including intestinal absorption disorders, systemic inflammation, and consumptive diseases. These

Corresponding author
Linglin Tian, tianlinlin587@163.com

pathophysiological changes could activate muscle protein degradation pathways, inhibit skeletal muscle protein synthesis, and cause muscle nutrition metabolism disorder. Weight loss and reduced physical activity may also be associated with sarcopenia.

Researchers have recently discovered that gut microbiota plays an important role in the development and progression of sarcopenia. Gut microbiota can affect hormone secretion and growth factor levels in the intestine through their metabolic products (such as short-chain fatty acids) and bile acids, affecting nutritional status and muscle reduction (*Zhao, Huang & Yu, 2021*). IBD is characterized by intestinal inflammation. This inflammation reduces intestinal absorption, causing malnutrition in patients. Furthermore, IBD patients often have a high degree of systemic inflammatory response, increasing overall energy consumption and reducing lean body tissue. Moreover, several key inflammatory mediators (such as TNF-$\alpha$ and IL-6) directly affect muscle metabolism, accelerating muscle loss and leading to sarcopenia (*Nucci et al., 2023*). The goal of this study was to review the epidemiological features of sarcopenia in patients with IBD and emphasize its impact on the prognosis of IBD.

## SURVEY METHODOLOGY

We searched PubMed, the Web of Science, Embase, Cochrane, the China National Knowledge Infrastructure, WanFang database and Chinese biomedical literature in English or Chinese by three searching strategies in the following: (1) "Inflammatory bowel disease/ Crohn's disease/ulcerative colitis" as subject heading or its free terms in combination with "sarcopenia" or its free terms; (2) "Inflammatory bowel disease/Crohn's disease /ulcerative colitis" or its free terms in combination with "Nutrition" or their free terms; (3) Keywords and subject headings related to "Inflammatory bowel disease/Crohn's disease/ulcerative colitis", "Sarcopenia/bioimpedance analysis/dual X-ray absorptiometry/skeletal muscle index" and "Nutrition/gut microbiota" were used. In addition, I would refer to these references of primary articles.

### Overview of sarcopenia
#### Definition of sarcopenia
Sarcopenia, also known as skeletal muscle loss, is a skeletal muscle-related disease characterized by the progressive and generalized loss of muscle mass and function. The term "sarcopenia" was coined by Irwin Rosenberg in 1988, and it was initially described as an age-related decline in lean tissue mass that affects mobility, nutritional status, and independence of individuals (*Cruz-Jentoft & Sayer, 2019*). Over the past several decades, researchers have continually updated their understanding of this disease. Two major developments have recently been made in the understanding of sarcopenia. First, six consensus definitions (*Cruz-Jentoft et al., 2010*; *Muscaritoli et al., 2010*; *Fielding et al., 2011*; *Morley et al., 2011*; *Chen et al., 2014*; *Studenski et al., 2014*) incorporating the concept of muscle function in addition to muscle mass have been established since 2010. This is a significant development as it has been shown that muscle function is more important than lean tissue mass decline in predicting clinically relevant outcomes. Second, sarcopenia was

officially recognized as an independent disease in 2016 (*Anker, Morley & von Haehling, 2016*).

The European Working Group on Sarcopenia in Older People (EWGSOP) categorizes the pathogenesis of sarcopenia into three stages (*Cruz-Jentoft et al., 2019*): (1) pre-sarcopenia, involving a decrease in the muscle mass only; (2) sarcopenia, characterized by a decline in skeletal muscle mass, strength, or function; and (3) severe sarcopenia, in which both strength and function are on the decline. These three stages are characterized by skeletal muscle fatigue or dysfunction. Acute sarcopenia is often observed in patients with acute illness or a sharp decrease in physical activity, whereas chronic sarcopenia is observed in patients with chronic inflammatory bowel disease (IBD), liver cirrhosis, and malignant tumors (*Cruz-Jentoft & Sayer, 2019*).

### Epidemiology of sarcopenia

Sarcopenia affects approximately 50 million people worldwide. Its incidence rate varies significantly across regions primarily due to the lack of standardized measurement methods and diagnostic thresholds. Another factor for this variation is the differences in the baseline characteristics of study populations globally. *Legrand et al. (2013)* conducted a study in Belgium in which 567 healthy participants aged $\geq$ 80 years were evaluated using the EWGSOP diagnostic criteria and bioimpedance analysis (BIA),they found that the occurrence rate of sarcopenia in the study population was 12.5%.A study conducted in Canada *Bouchard, Dionne & Brochu (2009)*, which involved 904 healthy elderly individuals and utilized dual X-ray absorptiometry (DXA) to assess skeletal muscle mass, found that the occurrence rate of sarcopenia was 12.5%. The study also revealed that males had a significantly higher prevalence rate of sarcopenia at 38.9%, compared to females at 17.8%. Another study (*Han et al., 2016*) based on the Asian Working Group on Sarcopenia criteria used BIA to measure the prevalence rate of sarcopenia among 1,069 individuals aged $\geq$ 60 years and discovered a significantly lower prevalence rate in Asia (4.1–11.5%) than in Europe and America. Moreover, geographical environment, dietary habits, lifestyle, and differences in genetic background can contribute to the regional differences in the incidence rate of sarcopenia.

### Assessment of sarcopenia

The diagnosis of sarcopenia includes three aspects: muscle strength, muscle mass, and physical performance. *Baumgartner et al. (1998)* used DXA to measure muscle loss in the limbs of elderly individuals; the diagnostic threshold for sarcopenia was an appendicular SMM (ASMM; kg)/height squared ($m^2$) value lower than two standard deviations (SDs) below the mean value of the young reference group. BIA revealed that individuals with lean body mass lower than two SDs below the mean value had a threefold higher risk of developing functional impairments than healthy individuals (*Janssen, Heymsfield & Ross, 2002*). The EWGSOP updated its definition of sarcopenia in 2010 by including low muscle mass and function. Two consensus articles on sarcopenia in the elderly were published in 2018: one updating EWGSOP2 (the 2018 update of EWGSOP) and the other discussing the International Clinical Practice Guidelines for Sarcopenia (ICFSR) for the management of

sarcopenia in the elderly. Both articles suggest that sarcopenia assessment should be based on the aforementioned three aspects.

*Imaging methods for assessing muscle mass.* Muscle mass can be assessed using DXA to evaluate the ASMM, BIA to assess the total body SMM (or limb muscle mass), and magnetic resonance imaging (MRI) or computed tomography (CT) to measure the cross-sectional muscle area of the lumbar spine. DXA can provide repeatable ASMM measurements within minutes using the same equipment and diagnostic thresholds. However, DXA equipment is not portable and cannot be widely used in community settings. Conversely, BIA is a low-cost, widely used, portable single-frequency device that can be adapted using muscle mass measurements from specific populations determined through DXA as a reference. However, BIA measurements are more likely to be influenced by the hydration status of patients (*Cruz-Jentoft et al., 2019*). MRI and CT are the gold standards for noninvasive assessment of muscle mass; however, their application is limited due to their high cost, low portability, and need for specially trained personnel. DXA, a noninvasive technique, is widely used to determine muscle mass.

The diagnostic thresholds for sarcopenia depend on the examination techniques and reference population used in the study. While quantifying muscle mass, absolute levels of SMM or ASMM should be adjusted for body parameters, including height squared (ASMM/height$^2$), weight (ASMM/weight), or body mass index (ASMM/BMI) (*Kim, Jang & Lim, 2016*). Several cohort studies have defined sarcopenia using the skeletal muscle index (SMI) as the ratio of the cross-sectional area of skeletal muscle at the third lumbar vertebra (L3) level determined *via* CT scan to height squared. Due to the variation in the currently used measurement methods, it is unclear whether the same method can be used to assess sarcopenia in all population groups, resulting in heterogeneity in sarcopenia assessment among IBD patients.

*Assessment of muscle strength and physical performance.* In addition to the assessment of body mass, the diagnosis of sarcopenia includes the following two aspects: (1) Muscle strength measurement using grip strength or chair stand tests, and (2) physical performance assessment using functional tests, such as gait speed, short physical performance battery, timed up-and-go, and 400-m walk tests, which are objective, measurable, exercise-related methods to assess overall body function (*Cruz-Jentoft & Sayer, 2019*).

## Sarcopenia and IBD
### Prevalence of Sarcopenia among patients with IBD
The nutritional characteristics of patients with IBD include weight loss during the active disease phase and gradual recovery during remission. Approximately 65–75% of patients with CD and 18–62% of patients with UC suffer from directly or indirectly induced by intestinal inflammation (*Malmstrom et al., 2016*). Current research on patients with IBD and sarcopenia shows that sarcopenia is more prevalent among IBD patients than those without IBD (Table 1).

However, there have been contrasting reports. A similar rate of skeletal muscle loss between acute trauma control and IBD patient groups has been observed (*Pedersen,*

**Table 1  Definitions and assessments of sarcopenia in IBD patients.**

| Studies | Type of study | Follow-up | Tools | Thresholds | IBD No. | | Prevalence | | |
|---|---|---|---|---|---|---|---|---|---|
| | | | | | UC | CD | In IBD | In UC | In CD |
| Bryant et al. (2013) | Prospectively, single centre. | 2012.04-2013.07 | ASMI ON DXA | ASMI ≤1 SD mean + HGS <mean | 42 | 95 | 21% (29/137) | 26% (11/42) | 19% (18/95) |
| Zhang et al. (2017a) | Retrospective, single centre | 2011.05-2014.03 | SMI at L3 (CT) | M: <55 cm$^2$/m$^2$ F: <39 cm$^2$/m$^2$ | – | 114 | 61% (70/114) | – | 61% (70/114) |
| Zhang et al. (2017b) | Retrospective, single centre. | 2010.06-2015.05 | SMI at L3 (CT) | M: <49.9 cm$^2$/m$^2$ F: <28.7 cm$^2$/m$^2$ | 99 | 105 | 44% (89/204) | 27% (27/99) | 59% (62/105) |
| Ding et al. (2017) | Prospectively, single centre. | 2007.01-2012.06 | SMI at L3 (CT) | M: <31.0 cm$^2$/m$^2$, F:<32.37 cm$^2$/m$^2$ | – | 106 | 25% (26/106) | – | 25% (26/106) |
| Cravo et al. (2017) | Retrospective, single centre | 2012-2015 | SMI at L3 (CT) | M:<43 cm$^2$/m$^2$ with BMI<25 or <53 cm$^2$/m$^2$ with BMI>25 kg/m$^2$ F: <41 cm$^2$/m$^2$ with BMI<25 | – | 71 | 31% (22/71) | – | 31% (22/71) |
| Bamba et al. (2017) | Retrospective, single centre | 2011.01-2016.12 | SMI at L3 (CT) | M: <42 cm$^2$/m$^2$ F: <38 cm$^2$/m$^2$ | 29 | 43 | 42% (30/72) | 48% (14/29) | 37% (16/43) |
| Adams et al. (2017) | Retrospective, single-center | 2012.08-2015.12 | SMI at L3 (CT) | M: <52.4 cm$^2$/m$^2$ F: <38.5 cm$^2$/m$^2$ | 14 | 76 | 46% (41/90) | 50% (7/14) | 45% (34/76) |
| Pedersen, Cromwell & Nau (2017) | Retrospective, singer centre | 2010-2015 | TPI on CT | TPI or HUAC <25% of each approach | 51 | 127 | 25% (44/178) | – | – |
| Thiberge et al. (2018) | Retrospective, singer centre | 2011.01-2015.12 | SMI at L3 (CT) | M: <55.4 cm2/m2, F:<38.9 cm2/m | – | 149 | 34% (50/149) | – | 34% (50/149) |
| Galata et al. (2020) | Retrospective, single centre | 2009.12-2017.12 | SMI at L3 (CT) or MRI | M: <41.5 cm2/m2, F: <31.8 cm2/m2 | – | 230 | 70% (162/230) | – | 70% (162/230) |
| Pizzoferrato et al. (2019) | Retrospective, single centre | 2017.02-2017.05 | ASMI on DEXA, SMI on BIA | M (D): <7.23 kg/m$^2$ F (D): <5.67 kg/m$^2$ M (B): <0.75 kg/m$^2$ F (B): <6.75 kg/m$^2$ | 58 | 69 | 36% (46/127) | – | – |
| Grillot et al. (2020) | Retrospective, single centre | 2010.01-2016.06 | SMI at L3 (CT) | M: <52.4 cm$^2$/m$^2$ F: <38.5 cm$^2$/m$^2$ | – | 88 | 58% (51/88) | – | 58% (51/88) |
| Lee et al. (2020) | Prospective, single centre | 2013.03-2017.03 | SMI at L3 (CT) | M: <49 cm$^2$/m$^2$, F: <31 cm$^2$/m$^2$ | – | 79 | 51% (40/79) | – | 51% (40/79) |
| Kang et al. (2020) | Retrospective, single centre | 2004.01-2017.12 | TPI on CT | M: <545 mm$^2$/m$^2$, F:<385 mm$^2$/m$^2$ | 169 | 274 | 35% (155/443) | – | – |
| Kurban et al. (2020) | Retrospective, single centre | – | FFMI on BIA | M: <17 kg/m$^2$, F: <15 kg/m$^2$ | | 47 | 85% (40/47) | – | 85% (40/47) |
| Berger et al. (2020) | Prospective, single centre | 2014.06-2016-04 | SMI at L3 (CT) | M: ≤43 cm$^2$/m$^2$ with BMI ≤25 F: ≤41 cm$^2$/m$^2$ | 32 | 59 | 45% (41/91) | 59% (19/32) | 37% (22/59) |
| Labarthe et al. (2020) | Retrospective, single centre | 2016.01-2019.01 | SMI at L3 (CT) | M: <54.4 cm$^2$/m$^2$, F: <38.9 cm$^2$/m$^2$. | – | 132 | 40% (53/132) | – | 40% (53/132) |
| Boparai et al. (2021) | Prospective, single centre | 2012.01-2019.12 | SMI at L3 (CT) | M: <36.5 cm$^2$/m$^2$, F: <30.2 cm$^2$/m$^2$ | – | 44 | 43% (19/44) | – | 43% (19/44) |
| Ünal et al. (2021) | Prospective, single centre | 2019.03-2019.-08 | HGS | HGS: M: <32 kg, F: <22 kg | 222 | 122 | 31% (107/344) | 32% (72/222) | 29% (35/122) |
| Bamba et al. (2021) | Retrospective multicentre | 2011.01-2018.08 | SMI at L3 (CT) | M: <42 cm$^2$/m$^2$ F: <38 cm$^2$/m$^2$ | 88 | 99 | 35% (65/187) | – | – |
| Campbell et al. (2022) | Retrospective, multicenter | 2005-2018 | PAI on CT | M: <4.64 cm$^2$/m$^2$ F: <4.05 cm$^2$/m$^2$ | – | 156 | 24%(38/156) | – | 24%(38/156) |
**Table 1** (*continued*)

| Studies | Type of study | Follow-up | Tools | Thresholds | IBD No. | | Prevalence | | |
|---|---|---|---|---|---|---|---|---|---|
| | | | | | UC | CD | In IBD | In UC | In CD |
| *Liu et al. (2022)* | Prospective, single-center | 2020.09-2021.09 | Handgrip, SMI on BIA | M: <7.0 kg/m$^2$ <br> F: <5.7 kg/m$^2$ | 85 | 25 | 50.8%(33/65) | – | – |
| *Ge et al. (2022)* | Prospective, single centre | 2015.12-2020.03 | SMI at L3 (CT) | M: <43.13 cm$^2$/m$^2$, <br> F: <37.81 cm$^2$/m$^2$ | 254 | – | 50% (127/254) | 50% (127/254) | – |
| *Nam et al. (2023)* | Retrospective, single-center | 1989.06-2016.12 | SMI at L3 (CT) | M: <49 cm$^2$/m$^2$ <br> F: <31 cm$^2$/m$^2$ | 173 | 854 | 56.8% (583/1,027) | 53.2% | 57.5% |

**Notes.**

M, Male; F, Female; ASMI, Appendicular skeletal muscle index (kg/m$^2$) =appendicular muscle mass (kg) divided by height squared (m$^2$); TPI, Total Psoas Index (cm$^2$/m$^2$)= total psoas muscle area (cm$^2$)/height (m2); SMI, Skeletal muscle index (cm$^2$/m$^2$) = skeletal muscle areas (cm$^2$)/height squared (m$^2$); FFMI, Fat-free mass index kg/m$^2$ = FFM (kg)/height$^2$ (m$^2$); HUAC, Hounsfield unit average calculation; BMI, Body mass index; HGS (kg), Handgrip-strength; CT, Computer tomography; MRI, Magnetic resonance imaging; DXA, Dual X-ray absorptiometry; BIA, Bioelectric impedance.

**Table 2  Sarcopenia prevalence in IBD classifications by area.**

| Area | No. of the studies in this area | Patient population | No. of Sarcopenia cases | Sarcopenia prevalence (%) |
|---|---|---|---|---|
| Asia | 12 | 2,915 | 1,268 | 43.5% |
| Europe | 6 | 797 | 384 | 48.2% |
| America | 4 | 515 | 164 | 31.8% |
| Oceania | 2 | 243 | 55 | 22.6% |

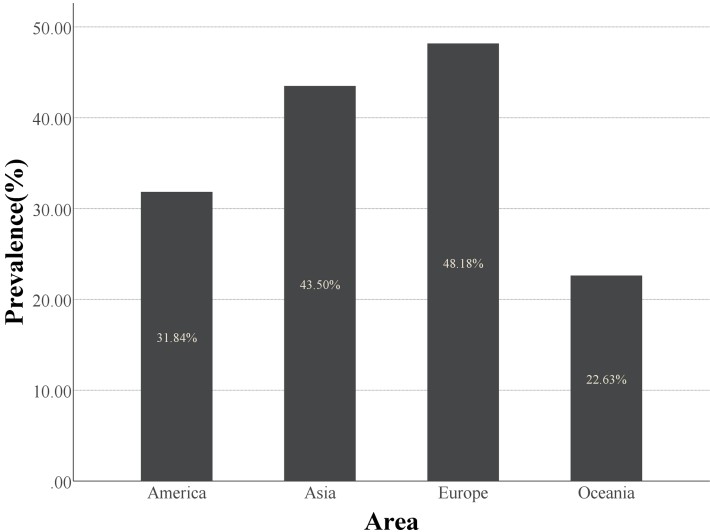

**Figure 1  The histogram of prevalence of sarcopenia in different regions.** Europe has a higher prevalence of sarcopenia than the other three regions.

*Cromwell & Nau, 2017*). This may be due to variations in the diagnosis method, ethnic groups, and IBD severity. The diagnostic methods for sarcopenia are still being developed, and obtaining more comprehensive and accurate prevalence data requires detailed multicenter, multiethnic disease severity stratification in prospective cohort studies. The definition of sarcopenia is influenced not only by the diagnostic threshold, but also by race (Table 2, Fig. 1).

Although CD and UC are classified as IBD, their disease characteristics are distinct. CD can develop in any part of the gastrointestinal tract, including the colon and small intestine, the primary nutrient absorption site. However, UC affects only the colon. Therefore, malnutrition plays a significant role in the pathogenesis of sarcopenia in patients with CD than in those with UC. Sarcopenia is prevalent in patients with CD, especially those requiring surgery. A retrospective study (*Zhang et al., 2017b*; *Zhang et al., 2017b*) discovered significantly higher skeletal muscle area (SMA), SMI, visceral fat area, and subcutaneous fat area in UC patients than in CD patients. The prevalence rate of sarcopenia was significantly lower in patients with UC (27.3%) than in patients with CD (59.0%), indicating that CD had a significant impact on nutritional status and body composition than UC. In a 90-day

prospective study, improved symptoms, changes in treatment plans, and the development of adverse outcomes (such as surgery, rehospitalization, and death) were observed in 110 patients with IBD (*Liu et al., 2022*). Their results also indicated that CD patients had lower SMM values than UC patients. Moreover, lack of physical activity contributed to muscle mass reduction in CD patients, as those in remission exhibited decreased muscle mass (*Werkstetter et al., 2012*).

However, three studies using DXA and BIA for body composition analysis discovered similar lean body masses between patients with CD and those with UC (*Capristo et al., 1998*; *Capristo et al., 1999*; *Jahnsen et al., 2003*). Another study using BIA for body composition assessment discovered no significant differences in body composition between CD and UC patients (*Bryant et al., 2013*). Several factors may have caused variation in the results. First, cohort sizes vary across studies and do not necessarily represent the overall population data. Second, disease duration, location, severity, and treatment methods vary across studies. Finally, the variation in body composition analysis techniques could also result in heterogeneous data collection.

### Sarcopenia as a predictor of clinical course and adverse events of IBD

*Sarcopenia is a manifestation of malnutrition for Patients with IBD.* Several blood biomarkers, including albumin, prealbumin, total protein, and BMI, are widely used in clinical practice for nutritional assessment and are considered standard diagnostic criteria for malnutrition. However, recent studies have shown that sarcopenia can develop in IBD patients with normal, reduced, or elevated BMI, indicating that sarcopenia can occur at any time in IBD patients and is not limited by BMI (*Adams et al., 2017*; *Ünal et al., 2021*). Several cohort studies investigating sarcopenia in individuals with inflammatory bowel disease (IBD) have indicated that gender, age, serum albumin levels, and body weight may play a role in determining the risk of developing sarcopenia. A recent systematic review of the outcomes of visceral fat and IBD revealed no significant difference between the BMI of patients with and without sarcopenia (*Berger et al., 2020*). A retrospective study discovered that 22% of IBD patients with a normal or elevated BMI and 36% with hypoalbuminemia had sarcopenia (*Campbell et al., 2022*). Although patients with low BMI and albumin levels are more likely to develop sarcopenia, these clinical indicators have no predictive or diagnostic value. A prospective study including 110 IBD patients suggested that low BMI and serum albumin levels are risk factors for sarcopenia, and these two easily accessible clinical indicators can help clinicians in the early diagnosis of sarcopenia in IBD patients (*Liu et al., 2022*).

Studies reported that the predictive value of BMI is limited to body fat assessment. Patients with high body fat and low muscle mass can still have normal BMI values. These findings highlight the importance of sarcopenia as an independent measurement indicator. Moreover, it can be inferred that using sarcopenia as a risk stratification criterion for disease severity positively affects the diagnosis and treatment of IBD patients.

*Association of Sarcopenia with the disease course and prognosis of patients with IBD.* Currently, over 15% of UC and 80% of CD patients undergo surgical treatment. The

effects of sarcopenia on the prediction of surgery and prognosis of patients with IBD are still being investigated by researchers.

Some studies (*Cravo et al., 2017*; *Kurban et al., 2020*; *Labarthe et al., 2020*; *Boparai et al., 2021*) show that the evaluation of body composition can be used as a marker of disease severity of patients with IBD. It can also be used as an independent predictor of IBD-related surgery.

Two studies (*Grillot et al., 2020*; *Ge et al., 2022*) clearly show that, the IBD patients with sarcopenia have a higher risk of surgery (OR 4.48, $p = 0.00$) (22.0% *vs* 7.1%, $p = 0.001$). Sarcopenia has a unique predictive function for surger $y$ (HR 0.318, CI [0.126–0.802], $p = 0.015$) (*Bamba et al., 2017*). In addition to the predictive value for surgery, patients with sarcopenia are more likely to fail in drug treatment (anti-TNF therapy) (OR 4.69, $P = 0.001$) and need rescue treatment more frequently (like colectomy) (*Ding et al., 2017*). The results of a study evaluating IBD patients who had started taking new antitumor necrosis factor (TNF) medications demonstrated that skeletal muscle loss could predict the likelihood of surgery in overweight patients ($P = 0.002$); however, it had no predictive value for clinical outcomes, such as hospitalization or medication changes (*Adams et al., 2017*). The intervals between readmissions in the control, pre-sarcopenia, and sarcopenia groups gradually decreased during the follow-up (89 *vs* 85 *vs* 57 days, $P < 0.001$) (*Liu et al., 2022*).

Surgical planning and risk assessment are essential components of the preoperative evaluation of IBD patients. Numerous comorbidities have been identified as risk factors for postoperative complications; however, these factors do not adequately measure the overall physical function and are not associated with the physiological reserve required for surgical recovery (*Hasselager & Gögenur, 2014*; *Erős et al., 2020*). Therefore, as an indicator of physiological reserve, muscle mass is a valuable factor that can help design treatment plans, determine surgical suitability, and predict outcomes (OR 9.24, $P = 0.04$) (*Zhang et al., 2017a*). Studies have shown a significant link between sarcopenia and intestinal resection in CD patients ($p = 0.01$), whereas no difference in the cumulative rate of surgery-free survival between sarcopenic and nonsarcopenic patients with UC ($p = 0.152$) (*Bamba et al., 2021*).

In addition, the relationship between sarcopenia and postoperative complications is also worth exploring. *Campbell et al. (2022)* observed that the proportion of patients with skeletal muscle loss is significantly higher in surgical cohorts (32%; mean PAI 5.29 $cm^2/m^2$) than in medical treatment cohorts (16%; mean PAI 5.86 $cm^2/m^2$)($P < 0.02$), indicating that these patients are more susceptible to malnutrition, cachexia, and subsequent loss of lean muscle mass. Sarcopenia is a potent independent predictor of major postoperative complications in CD patients, including transfusion, ICU admission, postoperative sepsis, and venous thromboembolism. These risks can be mitigated through preoperative interventions to improve the nutritional status of patients. A recent meta-analysis discovered that sarcopenia increased the risk of postoperative complications in patients with IBD by more than sixfold (*Erős et al., 2020*). There is significant correlation between IBD patients with sarcopenia and some infectious postoperative complications. For example, wound infection or dehiscence, anastomotic leak, abscess, sepsis, fistula,

pulmonary infection, and urinary tract infection (*Pedersen, Cromwell & Nau, 2017*; *Zhang et al., 2017a*; *Ge et al., 2022*), no significant association with non-infection complications (small bowel obstruction, ileus, deep vein thrombus, pulmonary embolism, bleeding, hematoma, or anemia) and sarcopenia (*Berger et al., 2020*). Clinicians can use SMI as an objective assessment tool to determine the surgical risk in CD patients and prevent complications related to preoperative nutritional management in sarcopenic patients with CD undergoing elective surgery (*Zhang et al., 2017a*).

In contrast, no association was between sarcopenia and surgery ($p = 0.066$), hospitalization ($p = 0.772$), or the initial use of new medications (biologics, $p = 0.481$; immunomodulators, $p = 0.320$; corticosteroid, $p = 0.842$) (*Lee et al., 2020*). Patients with skeletal muscle loss frequently underwent surgery, developed abscesses, and were re-hospitalized during the follow-up period. However, no correlation was identified between skeletal muscle loss and the use of new medication dosages ($p = 0.259$) (*Grillot et al., 2020*). Although subcutaneous fat and visceral fat are related to the poor outcome of IBD patients, sarcopenia does not have this effect (*Thiberge et al., 2018*). A retrospective study involving 1,027 IBD patients discovered no significant differences in the cumulative risks of using corticosteroids ($p = 0.073$), immunomodulators ($p = 0.116$), and biologics ($p = 0.743$) and undergoing intestinal resection ($p = 0.115$) during the follow-up period between patients with and without sarcopenia (*Nam et al., 2023*). In the univariate analysis, sarcopenia was a significant risk factor for perianal surgery in patients with CD, but multivariate logistic regression and Cox proportional hazard analyses revealed that sarcopenia is not a significant risk factor for perianal surgery (*Nam et al., 2023*). The heterogeneity in results can be attributed to various factors such as gender imbalance in this study, the impact of muscle loss diminishing over time, and this study only examining the short-term effects of muscle loss before and after diagnosis (Table 3).

In summary, sarcopenia may affect the prognosis of inflammatory bowel disease (IBD) in the following ways: (1) Slow recovery: Sarcopenia is related to the increased systemic inflammation levels in IBD patients, affecting the course of IBD; (2) Increased risk of complications: Some IBD patients require surgical treatment, and the risk of postoperative complications and infections increases in IBD patients with sarcopenia; (3) Decreased quality of life: IBD patients experience psychological stress, fatigue, and weakness, which may be related to sarcopenia; (4) Reduced treatment tolerance: IBD patients with sarcopenia have a reduced treatment response, a higher risk of adverse drug reactions, and difficulties with subsequent rehabilitation training (Fig. 2).

### Effects of clinical interventions for sarcopenia

Most current studies on the prognosis of sarcopenia are related to cancer. Cachexia-induced muscle decline is a multifactorial syndrome that cannot be reversed through traditional nutritional support and treatment (*Ryan et al., 2019*). However, muscle mass reduction due to aging or intestinal inflammation can be improved with medication or nutritional therapy. The anti-TNF-$\alpha$, anti-interleukin, anti-integrin, and Janus kinase (JAK) pathway inhibitors used to treat IBD can prevent skeletal muscle loss and block the catabolic process of skeletal muscle tissue, benefiting sarcopenic patients with IBD (*Nardone et al., 2021*).

Liu and Tian (2023), *PeerJ*, DOI 10.7717/peerj.16421

**Table 3  Summary of studies on sarcopenia in inflammatory bowel disease.**

| Studies | Lacation | Type of study | Number of patients (UC/CD) | Tools | Outcomes | Findings |
|---|---|---|---|---|---|---|
| *Bryant et al. (2013)* | Australia | Prospective, single center | 137 (42/95) | ASMI on DXA | Prevalence | Sarcopenia predicts osteopenia/osteoporosis of IBD. |
| *Zhang et al. (2017a)* | China | Retrospective, single center | 114 (0/114) | SMI at L3 (CT) | Surgical complications | Sarcopenia was a predictor of "major postoperative complications" |
| *Zhang et al. (2017b)* | China | Retrospective, surgery center, single center | 204 (99/105) | SMI at L3 (CT) | Disease activity; colectomy | Sarcopenia was associated with higher disease activity and poor outcomes in UC. |
| *Ding et al. (2017)* | Australia | Prospective, single center | 106 (0/106) | SMI at L3 (CT) | Biologic initiation. | Myopenia was associated with primary nonresponse of anti-TNF therapy. |
| *Cravo et al. (2017)* | Portugal | Retrospective, single center | 71 (0/71) | SMI at L3 (CT) | Severe phenotypes | A lower muscle attenuation and a high visceral fat index suggest a more serious phenotype. |
| *Bamba et al. (2017)* | Japan | Retrospective, single center | 72 (29/43) | SMI at L3 (CT) | Skeletal muscle volume and prognosis of patients. | Sarcopenia predicted the need for intestinal resection. |
| *Adams et al. (2017)* | USA | Retrospective, single center | 90 (14/76) | SMI at L3 (CT) | Identify malnutrition; Surgery;biologic initiation | Sarcopenia was a predictor of surgery in those with BMI >25. |
| *Pedersen, Cromwell & Nau (2017)* | USA | Retrospective, single center | 178 (51/127) | TPI or HUAC on CT | Surgical complications | In patients <40 years, sarcopenia is associated with surgical complications. |
| *Thiberge et al. (2018)* | France | Retrospective, single center | 149 (0/149) | SMI at L3 (CT) | Disease severity; Surgical complication | Subcutaneous and visceral adiposity correlated inversely with adverse outcomes |
| *Galata et al. (2020)* | Germany | Retrospective, single center | 230 (0/230) | SMI at L3 (CT)or MRI | Surgical complication | SMI could stratify the risk of postoperative complications. |
| *Pizzoferrato et al. (2019)* | Italy | Retrospective, multicenter | 127 (58/69) | ASMI on DEXA/ SMI on BIA | Asthenia degree; Quality of life | Sarcopenia brings fatigue perception and reduces quality of life. |

Liu and Tian (2023), *PeerJ*, DOI 10.7717/peerj.16421

**Table 3** (*continued*)

| Studies | Lacation | Type of study | Number of patients (UC/CD) | Tools | Outcomes | Findings |
|---|---|---|---|---|---|---|
| *Grillot et al. (2020)* | France | Retrospective, single center | 88 (0/88) | SMI at L3 (CT) | Abscesses; Hospitalizations; Digestive surgery | Sarcopenia and visceral obesity were associated with adverse outcomes. |
| *Lee et al. (2020)* | South Korea | Prospective, single center | 79 (0/79) | SMI at L3 (CT) | Hospitalization; Surgery; Steroids; Immunomodulators; Biologics | No association between sarcopenia and outcomes. |
| *Kang et al. (2020)* | South Korea | Retrospective, single center | 433 (169/274) | TPI on CT | non-alcoholic fatty liver disease | Rates of NAFLD are higher in sarcopenic IBD patients. |
| *Kurban et al. (2020)* | South Korea | Retrospective, single center | 47 (0/47) | FFMI on BIA | Nutritional status | Sarcopenia could be used in the assessment of disease severity in CD patients. |
| *Berger et al. (2020)* | USA | Prospective, single center | 91 (32/59) | SMI at L3 (CT) | Postoperative complications. | Sarcopenia could predict 30-day postoperative infection complications. |
| *Labarthe et al. (2020)* | France | Retrospective, single center | 132 (0/132) | SMI at L3 (CT) | Disease activity | Body composition is related to disease activity. |
| *Boparai et al. (2021)* | India | Prospective, single center | 44 (0/44) | SMI at L3 (CT) | Surgery; | Combination of sarcopenia and high visceral fat predict worse outcomes. |
| *Ünal et al. (2021)* | Turkey | Prospective, single center | 344 (222/122) | HGS | Clinical remission | Body composition analysis can prevent sarcopenia-related poor outcomes in remission. |
| *Bamba et al. (2021)* | Japan | Retrospective, multicenter | 187 (88/99) | SMI at L3 (CT) | Prolonged length of stay; Intestinal resection | Sarcopenia brings prolonged length of stay; Muscle volume and visceral adipose tissue volume are associated with intestinal resection. |
| *Campbell et al. (2022)* | USA | Retrospective, multicenter | 156 (0/156) | PAI on CT | New biologics; Surgeries | Sarcopenia predicts Progression to Surgery Among Medically Treated Patients. |

Liu and Tian (2023), *PeerJ*, DOI 10.7717/peerj.16421

**Table 3** (*continued*)

| Studies | Lacation | Type of study | Number of patients (UC/CD) | Tools | Outcomes | Findings |
|---|---|---|---|---|---|---|
| *Liu et al. (2022)* | China | Prospective, single center | 110 (85/25) | Handgrip, SMI on BIA | Surgery; Re-hospitalization | Sarcopenia and sarcopenic obesity are associated with poor outcomes. |
| *Ge et al. (2022)* | China | Prospective, single center | 254 (254/0) | SMI at L3 (CT) | ASUC; Colectomy | Sarcopenia predicts the need for colectomy in ASUC. |
| *Nam et al. (2023)* | Korea | Retrospective, single center | 1027 (173/854) | SMI at L3 (CT) | Steroids; Immunomodulators; Biologics; Bowel resections | Sarcopenia have no significant prognostic value for medical treatment and bowel resection. |

**Notes.**
ASUC: Acute Severe Ulcerative Colitis.
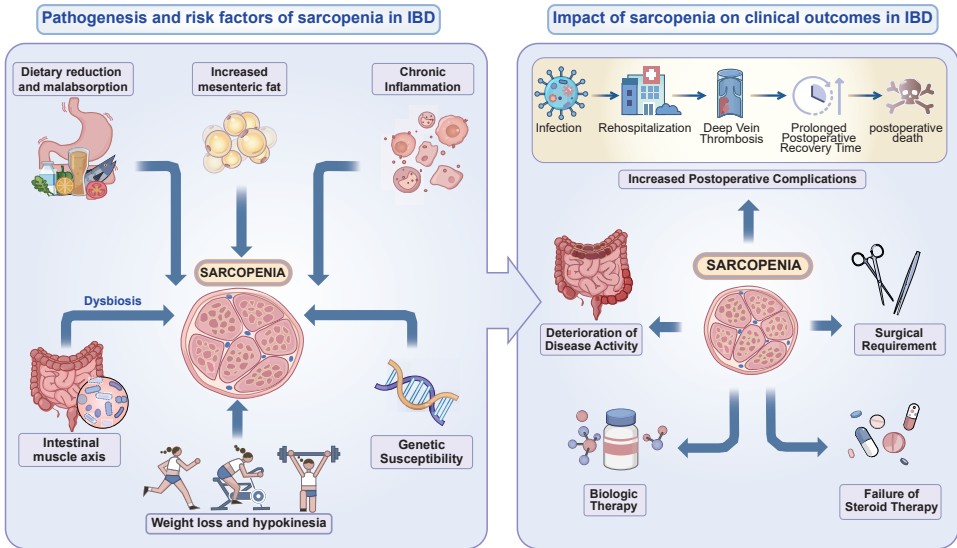

**Figure 2** The pathogenesis and risk factors of sarcopenia in patients with IBD and impact of sarcopenia on clinical outcomes in IBD.

Colonic resection in patients with CD or UC can induce the recovery of skeletal muscle volume, resulting in a progressive increase in muscle volume and mass (*Subramaniam et al., 2015*). Few studies have reported improvements in BMI and muscle parameters after three months of infliximab or adalimumab treatment. Therefore, it can be inferred that anti-TNF-$\alpha$ therapy positively affects the nutritional status and body composition of IBD patients (*Csontos et al., 2016*), which may be associated with a reduction in the inflammatory burden following drug treatment.

## CONCLUSION AND OUTLOOK

The results indicate that sarcopenia is an effective indicator of the nutritional status, body composition and prognosis of IBD patients. By identifying various clinical indicators such as serum albumin level and BMI, the risk of skeletal muscle loss can be detected at an early stage. This allows for timely determination of surgical intervention in patients with IBD, leading to a decrease in the occurrence of unfavorable outcomes and an enhancement in the quality of life for these individuals. However, no reliable, independent predictors have been identified for sarcopenia. Relying solely on imaging identification presents challenges in detecting sarcopenia during its initial stages, often leading to delayed diagnosis until there is a substantial decline in muscle mass and function, resulting in significantly weakened physical performance. In order to effectively manage sarcopenia in patients with IBD, it is vital to identify severe cases early on and predict prognosis and surgical outcomes. This highlights the importance of incorporating muscle mass assessment as a regular evaluation criterion for individuals with IBD. Additionally, the determination of diagnosis and threshold values for sarcopenia is currently uncertain due to discrepancies in measurement techniques. Moreover, biological agents can improve muscle mass reduction and inhibit

intestinal inflammation. Therefore, conducting prospective studies on this topic can benefit sarcopenic patients with IBD.

Future research can be expanded through the following aspects: (1) Analyzing the molecular mechanisms of sarcopenia pathogenesis in IBD and searching for effective therapeutic targets; (2) Investigating biomarkers with independent predictive value for diagnosing sarcopenia in IBD patients; (3) Identifying the optimal diagnostic tools for sarcopenia in IBD patients through multicenter, prospective studies; (4) Developing drug treatment strategies for sarcopenia in IBD patients.

### Funding
This work was supported by the General Project of Shanxi Natural Science Foundation (202103021224397). Linglin Tain provided financial support for the conduct of the research. The funders had no role in study design, data collection and analysis, decision to publish, or preparation of the manuscript.

### Grant Disclosures
The following grant information was disclosed by the authors:
General Project of Shanxi Natural Science Foundation: 202103021224397.

### Competing Interests
The authors declare there are no competing interests.

### Author Contributions

- Yang Liu performed the experiments, analyzed the data, prepared figures and/or tables, authored or reviewed drafts of the article, and approved the final draft.
- Linglin Tian conceived and designed the experiments, performed the experiments, prepared figures and/or tables, authored or reviewed drafts of the article, and approved the final draft.

### Data Availability
This is a literature review.

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
