# Peer review of "Research progress on the predictive role of sarcopenia in the course and prognosis of inflammatory bowel disease"

_PeerJ, doi:10.7717/peerj.16421_

## Round 0.1 · original submission · Major Revisions

1. Please improve the writing and formatting style of the article overall.
2. Use schematic to make the article more comprehensive.
3. Try to maintain the flow of the writing with good connections from one section to the next sections.
4. Add table with the common risk factors for sarcopenia and IBD with proper references.

Reviewer 1 ·

Basic reporting

The authors have talked about sarcopenia and inflammatory bowel disease. authors try to understand the correlation between sarcopenia and inflammatory bowel disease.

Experimental design

no comment

Validity of the findings

Authors can still use better information and literature to put their point of view.

Additional comments

Please, improve the quality of the manuscript by rewriting the manuscript.

·

Basic reporting

I read carefully the research manuscript entitled Research Progress on the predictive role of Sarcopenia in the Course and prognosis of inflammatory bowel disease.' This research is generally interesting however it needs to address a few comments, strong interpretation, and table rectification thus requiring substantial minor revision.

1. Authors should modify keywords. Please mention appropriate keywords and also rearrange keywords alphabetically that support your manuscripts.
2. Authors should be concerned with the references mentioned in the manuscript.
3. Authors should adjust spacing throughout the manuscript.
4. Authors should rearrange all tables in the manuscript. Please put an extra column on the right side of the table and put references accordingly.
5. Authors should remove the study column from the table. Kindly add more information that strongly supports your hypothesis.

Experimental design

1. Authors should incorporate a few graphical representations based on the table’s data that would be very useful for better understanding.
2. Authors should clearly mention the relationship between sarcopenia and IBD. Try to mention a graphical representation that supports a clear view of the prognostic role of sarcopenia in the course and prognosis of IBD.
3. Authors should rectify the references style in the manuscript.
4. Authors should incorporate a few statistical data from previously published that would support statistical evidence of this study.

Validity of the findings

Authors should work to demonstrate clear findings from this study.

Additional comments

Authors should work to rearrange all tables, including a flow chart, a few statistical data, and a graphical representation that strongly support your hypothesis.

---

## Round 0.2 · accepted · Accept

Thank you for addressing the comments from the reviewers.

·

Basic reporting

The authors have changed all the points that I have mentioned in my previous comments. Now the manuscript is much better and compact with the most valuable aspects of IBD.

Experimental design

The survey methodology sections are more organized than the previous version.

Validity of the findings

The Authors have mentioned the future directions and investigations of the IBD which will be more effective is this research field.

Additional comments

No comments